# Integrated Analysis of Transcriptome, microRNAs, and Chromatin Accessibility Revealed Potential Early B-Cell Factor1-Regulated Transcriptional Networks during the Early Development of Fetal Brown Adipose Tissues in Rabbits

**DOI:** 10.3390/cells11172675

**Published:** 2022-08-28

**Authors:** Kun Du, Yu Shi, Xue Bai, Li Chen, Wenqiang Sun, Shiyi Chen, Jie Wang, Xianbo Jia, Songjia Lai

**Affiliations:** Farm Animal Genetic Resources Exploration and Innovation Key Laboratory of Sichuan Province, Sichuan Agricultural University, Chengdu 611130, China

**Keywords:** BAT, fetal BAT, rabbits, ATAC-seq, epigenetics, miRNAs

## Abstract

In domestic mammals, cold stress decreases the survival rate of newborns and increases the cost of management. Brown adipose tissue (BAT) is the main thermogenic organ for cubs, and well-developed fetal BAT (FBAT) is beneficial for newborns to maintain core temperatures during the first several days of life. However, our knowledge of the epigenetic mechanisms during the early development of FBAT remains largely unknown. Rabbits (*Oryctolagus cuniculus*) are economically important domestic animals. In this study, a histological analysis shows that the tissue content, thermogenic capacity, and lipid content of FBAT dramatically increases from gestational day 21 (G21) to gestational day 24 (G24) in rabbits. RNA-seq, microRNA-seq (miRNA-seq), and the assay for transposase-accessible chromatin with high-throughput sequencing (ATAC-seq) show that many genes, miRNAs, and chromatin-accessible regions (referred to as peaks) were identified as significantly changed from G21 to G24, respectively. The upregulated genes from G21 to G24 were significantly enriched in the mitochondrial metabolism and thermogenesis-related signal pathways. The integrated analysis of transcriptome and chromatin accessibility reveals that the peaks in the promoters have a more regulatory effect than peaks in other genomic elements on the expression of their nearby genes in FBATs. The upregulated genes that are associated with increased chromatin accessibility in the promoter regions are involved in the energy metabolism-related signaling pathways. The genes that have a greater tendency to be regulated by miRNAs than the chromatin accessibility in gene promoters are involved in the apelin, insulin, and endocytosis signaling pathways. Furthermore, genome-wide transcription factor (TF) footprinting analysis identifies early B-cell factor1 (EBF1) as playing a key role during early FBAT development. The carbon metabolism, citrate cycle, and PPAR signaling pathways are significantly enriched by the predicted EBF1-regulated cascade TF-network. In conclusion, our work provides a framework for understanding epigenetics regulatory mechanisms underlying the early development of FBAT and identifies potential TF involved in the early development of FBAT in rabbits.

## 1. Introduction

The peak time period for the loss of newborn domestic mammals is from the time of birth through the first several days of life [1,2], during which cold stress is common for domestic mammals. Hypothermia caused by cold stress increases morbidity and mortality and decreases the survival rate of newborns [3,4,5,6,7]. Newborn cubs must quickly adapt to an extrauterine low-temperature environment to maintain their thermal homeostasis. In the husbandry industry, the additional heat preservation facilities and supplemental heat increase the cost of management of infant mammals [8]. To decrease the mortality and the management cost for cubs, the thermogenic capacity of animals has been considered an important trait in the emerging breeding programs of domestic mammals [4,9,10].

Brown adipose tissue (BAT) is the main thermogenic organ for newborns, which dissipates chemical energy as heat to protect animals from hypothermia [11]. The non-shivering heat of BAT, being mediated by the expressed uncoupling protein 1 (UCP1), compensates for the insufficient heat generation of muscles in newborns [11,12,13,14,15]. Because of the highly efficient calorie consumption, BAT is also a target for treating human obesity and regulating the total fat deposition of domestic animals [4,16]. The weight of BAT relative to body weight reaches a maximum at birth and falls after birth, suggesting the importance of the development of intrauterine fetal BAT (FBAT) for newborn animals [17,18]. On the other hand, well-developed FBAT is beneficial for maintaining core temperature when suffering extrauterine low temperature at birth. Therefore, it is of great significance to elucidate the mechanisms controlling FBAT development in domestic animals.

In humans and mice, a network of transcription factors (TFs), such as Early B-cell factor2 (EBF2), peroxisome proliferative activated receptor-gamma (PPARG), and nuclear factor I-A (NFIA) were identified as controlling brown adipocyte commitment or differentiation [19]. During the past decade, studies showed that the BAT gene program is also regulated by epigenetic controls, such as non-coding RNA (e.g., microRNAs) regulation [20], histone modifications [21,22], and RNA methylation [23]. Recent exciting progress demonstrated that the expression of the BAT gene was associated with chromatin accessibility [24,25,26]. The epigenetic machinery of chromatin accessibility enabled DNA binding proteins (e.g., TFs) to physically interact with DNA sequences, thereby regulating the transcription of bound genes [27]. Maintaining the chromatin accessibility of the binding sites of key genes was found to be essential for the thermogenesis of BAT [24,25]. In domestic animals, recent transcriptomic studies have revealed the RNA molecules regulating the postpartum development of BAT in goats [28], cattle [16], and rabbits [29]. Nevertheless, our knowledge of the regulatory mechanisms of epigenetic regulation during FBAT development, especially early FBAT development, remains largely unknown in domestic mammals.

Rabbits (*Oryctolagus cuniculus*) are economically important domestic animals. Despite the low body fat-rate of adults, the newborn rabbits have considerable BAT mass at birth (ca. 5% of total weight), which is mainly distributed in the interscapular and cervical regions [17]. The BATs are essential for maintaining the core temperature of newborn rabbits [30]. Given the large size and easy accessibility of FBAT, rabbits also provide an ideal animal model for investigating the development of FBAT during the prenatal period. In the present study, histological assays are firstly carried out to characterize the FBAT in the early perinatal period of rabbits. Then, RNA-seq, microRNA-seq, and assay for transposase-accessible chromatin with high-throughput sequencing (ATAC-seq) are carried out to detect the changes of the transcriptome, microRNAs, and chromatin accessibility, respectively. Finally, an integrated analysis reveals the key role of the early B-cell factor1 (EBF1) in promoting early FBAT development. Our works are expected to shed light on the molecular mechanisms regulating early FBAT development and provide the data for future research on the BAT-based thermogenic capacity of domestic animals.

## 2. Materials and Methods

### 2.1. Ethics Approval

All surgical procedures involving rabbits were performed according to the approved protocols of the Biological Studies Animal Care and Use Committee, Sichuan Province, China. Rabbits had free access to food and water under normal conditions and were humanely sacrificed as necessary to ameliorate suffering (Ethics Code: DKY2020102011).

### 2.2. Experimental Animals and Histological Observation

In this study, female Tianfu Black rabbits (native species in Sichuan province of China) that were analogous in age, body weight, and parity were raised at the breeding center of Sichuan Agricultural University, Ya’an, China. These rabbits were fed a standard diet, as described in our previous study, and water ad libitum [31]. The female rabbits were fertilized at the same time. FBATs were collected from interscapular regions of fetal rabbits at gestational day 21 (G21) and gestational day 24 (G24). The FBATs were fixed for 24 h using 4% formaldehyde at room temperature (RT). Then, the FBATs were dehydrated through a graded ethanol series, transferred to xylene, and embedded in paraffin wax. The paraffin wax slices of the FBATs were stained using hematoxylin-eosin (H&E). The primary UCP1 antibody (anti-UCP1 rabbit polyclonal antibody, Sangon Biotech, Shanghai, China) was used for IF (immunofluorescence) and immunohistochemistry (IHC) assays. The slices were incubated in the primary UCP1 antibody (1:100) at RT for 1 h and washed three times using phosphate buffer saline (PBS). The slices used in the IF assay were incubated in the fluorescein isothiocyanate-conjugated secondary antibody (1:500, BOSTER, Wuhan, China) for 1 h and washed three times using PBS. The slices used in the IHC assay were incubated in horseradish peroxidase-marked secondary antibody (1:500, Sangon Biotech, Shanghai, China) for 1 h and washed three times using PBS. All slices involved in H&E staining, IF, and IHC were observed using an Olympus BX-50F light microscope (Olympus BX53, Tokyo, Japan) (IF slices were observed under a 510 nm wavelength fluorescent light source).

### 2.3. Library Preparation, Sequencing, and Analysis of RNA-Seq

The FBATs of 4 offspring from one mother were pooled as one sample. The FBATs collected from 3 mothers at G21 (*n* = 3) and 3 mothers at G24 (*n* = 3) were snap-frozen in liquid nitrogen and stored at −80 °C until RNA isolation. The total RNA was extracted from FBATs using TRIzol Reagent (Thermo Fisher, Shanghai, China). DNA digestion was carried out after RNA extraction by using DNaseI (Sigma-Aldrich, St. Louis, Missouri, USA). RNA quality was determined by examining A260/A280 with Nanodrop2000 (Thermo Fisher, Shanghai, China). RNA integrity was confirmed by 1.5% agarose gel electrophoresis. The qualified RNAs were finally quantified by Qubit3.0 with a QubitTM RNA Broad Range Assay kit (Life Technologies, Carlsbad, USA). The RNA samples that had concentration > 200 ng/μL, value of OD260/280 ranging from 1.8 to 2.2, OD260/230 > 2.0, RNA integrity number (RIN) > 7 were used for downstream library construction. A total of 2 μg RNA was used to construct a stranded RNA-sequencing library using the Ribo-off rRNA Depletion Kit (Illumina, California, USA) and KC^TM^ Stranded mRNA Library Prep Kit for Illumina^®^ (Seqhealth Co., Ltd., Wuhan, China), following the manufacturer’s instructions. The 200–500 bp segments in the library products were then enriched, quantified, and sequenced on an Illumina NovaSeq 6000 platform. Finally, paired-end 150 bp reads were generated.

The low-quality reads of raw reads were discarded, and the reads contaminated with adaptor sequences of raw reads were trimmed using Trimmomatic software [32]. Clean reads were mapped back to the rabbit genome using Hisat2 [33]. Read counts were estimated and normalized using Stringtie software [34]. Differentially expressed genes (DEGs) were identified using DEseq2 [35] with thresholds of |log2(fold-change)| > 1 and Padj < 0.01.

### 2.4. Library Preparation, Sequencing, and Analysis of miRNA-Seq

The methods of sample preparation and RNA isolation were performed as described in RNA-seq library preparation. The quantified RNA sample was used for miRNA library preparation using the KC-Digital^TM^ small RNA Library Prep Kit for Illumina^®^ (Seqhealth Co., Ltd. Wuhan, China) following the manufacturer’s instructions. The eluted cDNA library was separated by 6% PAGE gel. The approximately 160 bp bands were isolated, purified, and quantified by Qubit3.0, and finally sequenced on an Illumina Hiseq X-10 sequencer.

Raw sequencing reads were firstly filtered by the Fastx-toolkit (http://hannonlab.cshl.edu/fastx_toolkit/, accessed on 8 February 2022) for discarding of the low-quality reads, and then the adaptor sequences of raw data were trimmed by Cutadapt [36]. The clean reads were mapped to known primary-miRNA in the database of mirBase [37] to classify the known and novel miRNAs based on the secondary structure, Dicer enzyme cleavage site, and minimum free energy indexes using MirDeep2 [38]. The normalization of the miRNA data (CPM, counts per million reads) and differential analysis of miRNA was conducted using DEseq2 [35]. The miRNA with thresholds of |log2(fold-change)| > 1 and *p*-value < 0.05 were considered differentially expressed miRNAs (DEmiRNAs). The post-transcriptional repression through miRNA seed region binding to the 3′ untranslated region (UTR) of target mRNA was considered as the canonical mode of miRNA-mediated gene regulation [39]. In this study, the 3′ UTR sequences of mRNAs were used to predict the target genes of miRNAs using the miRanda [40] with the thresholds of score > 150 and RNA hybrid energy < −15 and the RNAhybrid [41] with the thresholds of RNA hybrid energy < −15 and RNA hybrid *p*-value < 0.05.

### 2.5. Library Preparation, Sequencing, and Analysis of ATAC-Seq

The method of sample preparation was performed as described in RNA-seq library preparation. The samples collected from 3 mothers at G21 (*n* = 3) and 4 mothers at G24 (*n* = 4) were snap-frozen in liquid nitrogen and stored at −80 °C until cell isolation. The ATAC-seq library was constructed using the methods previously described [42]. Briefly, a total of approximately 50,000 cells were dissociated from each pooled sample to obtain single-cell suspensions. Then, the cells were added to nuclear isolation buffer and washed repeatedly using a nuclear wash buffer following a standard nuclear isolation protocol. The purified nuclei were then mixed with Tn5 transposase carrying sequencing adapters and incubated for a half-hour at 37 °C. The transposed DNA fragments were then purified and amplified using polymerase chain reaction (PCR). The constructed ATAC-seq libraries were purified using a Qiagen MinElute PCR Purification Kit (QIAGEN, Dusseldorf, Germany) following the manufacturer’s instructions. Finally, the qualified libraries were sequenced on an Illumina NovaSeq 6000 platform and paired-end reads were generated.

The adapter sequences and low-quality reads were removed using Trimmomatic software [32]. Clean reads were then mapped back to the rabbit genome (OryCun2.0, ENSEMBL release 101) using Bowtie2 software [43]. The PCR duplicates were removed using the MarkDuplicates program in Picard [44]. MACS2 was used to call the chromatin regions (ATAC-seq peaks) with the q-value 0.05 cutoff [45]. A consensus peak-set of the seven samples was merged using DiffBind [46]. The raw read counts of peaks were normalized using the CPM method in DiffBind, which was calculated by dividing the raw read counts by the count number of reads that mapped to all peaks [46]. The genome-wide ATAC-seq peaks were annotated using CHIPSeeker [47]. Differential peaks were identified by DESeq2 [35] using the raw read counts of the peaks, with the thresholds of |log2(fold-change)| > 1.5 and Padj < 0.05. For genes containing both increased and decreased peaks in a promoter region (±3 kb of TSSs), the peak that had the most increased or decreased degree was considered the major peak in a gene promoter in the promoter-based chromatin accessibility analysis. The sequences of differential peaks were subjected to TF binding motif enrichment using HOMER [48]. To further investigate the TF binding events, we downloaded 765 vertebrate TF binding motifs from JASPAR [49]. Then, TOBIAS [50] was used to correct the cutting bias of Tn5 transposase and TF binding detection. As described in TOBIAS [50], all TFs with -log10(*p*-value) above the 95% quantile or differential binding scores smaller/larger than the 5% and 95% quantiles (top 5% in each direction) were considered differential binding TFs.

### 2.6. Functional Annotation and Pathway Enrichment

Gene Ontology (GO) enrichment and Kyoto Encyclopedia of Genes and Genomes (KEGG) pathway analysis was performed using Clusterprofiler [51]. The enriched GO term or KEGG pathway with a *p*-value < 0.05 was considered significant.

### 2.7. Quantitative Real-Time PCR (RT-qPCR)

The RT-qPCR primers of selected genes were designed using Primer-blast (https://www.ncbi.nlm.nih.gov/tools/primer-blast/, accessed on 8 February 2022) and listed in Appendix A. For protein-coding genes (PCGs) validation, the gDNA of approximately 1 μg total RNA was removed using gDNA eraser (Foregene, Chengdu China). The RNA that gDNA had removed was reversed and transcribed to cDNA using Master Premix For qPCR RT Easy^TM^ II (Foregene, Chengdu, China) according to the manufacturer’s instruction. The cDNA was then used as the amplification template for the two-step PCR reaction. The PCR reaction was performed on an CFX96^TM^ Real-Time PCR Detection System (BioRad, Heracles, CA, USA) using the SYBR Premix Ex Taq^TM^ II (Novoprotein, Jiangsu, China). The PCR reactions were performed under the following conditions: pre-denaturation at 95 *°C* for 30 s, followed by 40 cycles of denaturation at 95 °C for 15 s and annealing/extension at 58.8 °C for 20 s. Melting curve analysis was performed from 55 to 95 °C with increments of 0.5 °C, and the unique melting peak of the expected PCR product was used to validate the primer specificity. Two technical replicates were set for one individual experimental replicate. The Ct values of target genes were normalized to the *RN18S* gene using 2^−ΔΔCt^ method.

For miRNA validation, approximately 3 μg RNA was reversed and transcribed to the cDNA of miRNA using the Mir-X miRNA First-Strand Synthesis Kit (Takara, Dalian, China). According to the manufacturer’s instruction of the Mir-X miRNA First-Strand Synthesis Kit, the 5′ miRNA-specific primers were designed using mature sequences of miRNAs (U bases were replaced using T bases), and the 3′ primers were used mRQ 3′ Primer in the reagent. The SYBR Premix Ex Taq^TM^ II (Novoprotein, Jiangsu, China) was used in the PCR reaction. The PCR reactions were performed under the following conditions: pre-denaturation at 95 °C for 30 s, followed by 40 cycles of denaturation at 95 °C for 15 s and annealing/extension at 58.8 °C for 20 s. Melting curve analysis was performed from 55 to 95 °C with increments of 0.5 °C, and the unique melting peak of the expected PCR product was used to validate the primer specificity. The Ct values of target miRNAs were normalized to the *U6* gene using 2^−ΔΔCt^ method.

### 2.8. Statistical Analysis

Statistical analyses, including t-test and one-way ANOVA, were conducted on R software. The *p*-value < 0.05 was considered significant.

## 3. Results

### 3.1. Changes of Histological Characterization and Genome-Wide Gene Expression during the Early FBATs Development in Rabbits

In this study, the FBATs were collected from interscapular regions of rabbits at gestational day 21 (G21) and gestational day 24 (G24). Morphological observations showed that there are trace amounts of FBATs in the interscapular region at G21. The FBATs dramatically increased and expanded to the dorsum at G24 (Figure 1A). There were obvious histological changes from G21 to G24. The H&E staining showed that cell size, lipid content, and the number of mature brown adipocytes (multilocular adipocytes) increased from G21 to G24 (Figure 1B). The RT-qPCR assay showed that the gene expression levels of the *UCP1* were significantly upregulated by approximately 700 folds from D21 to D24 (*p*-value < 0.01). The thermogenic genes *CIDEA* and *PPARGCIA* and the mitochondrial genes *COX1*, *COX2*, *ND1*, and *ND2* were also significantly upregulated from G21 to G24 (*p*-value < 0.01, Figure 1C). To further determine the development of FBATs, we detected the content of the UCP1 protein using IF and IHC assays. Both the results of IF (Figure 1D) and IHC (Figure 1E) show that the number of UCP1+ cells increased from G21 to G24. These results indicate that the thermogenic capacity and lipid content of FBATs increases from G21 to G24.

To determine global gene expression changes from G21 to G24, samples from each group (*n* = three per group) were subjected to deep RNA-seq. An average of 118.42 million clean reads (paired-end 150 bp) and an 88.15% mapping ratio were obtained from our deep RNA-seq libraries (Appendix A). Based on gene expression levels, hierarchical clustering sorted the samples into two distinct clusters corresponding to G21 and G24 (Figure 2A). Differential analysis identified 1618 and 1198 significantly up- and downregulated genes (Figure 2B, Appendix A). These data reveal transcriptome changes during the early development of FBATs. The top 10 significantly enriched gene ontology-biological process (GO-BP) terms by the upregulated genes from G21 to G24 were all associated with energy metabolisms, such as the top 3 enriched GO-BP terms of cellular respiration, generation of precursor metabolites and energy, and energy derivation by oxidation of organic compounds (Figure 2C). As expected, lipid metabolism-related GO-BP terms were also significantly enriched by the upregulated genes, such as lipid metabolic process, fatty acid metabolic process, and fatty acid oxidation (Figure 2C). The KEGG pathway analysis of the upregulated genes showed that mitochondrial metabolism and thermogenesis-associated pathways were significantly enriched, such as pathways of carbon metabolism, citrate cycle, thermogenesis, PPAR signaling pathway, oxidative phosphorylation, and fatty acid metabolism (Figure 2C). The top 10 significantly enriched GO-BP terms by the downregulated genes from G21 to G24 were associated with cell proliferation, such as the top 3 enriched GO-BP terms of cell cycle process, cell cycle, and chromosome segregation (Figure 2D). The downregulated genes were enriched in the KEGG pathways associated with the cell proliferation process, such as the pathways of DNA replication, cell cycle, and nucleotide excision repair (Figure 2D).

### 3.2. Identification of miRNAs during Early FBAT Development

To investigate the post-transcriptional epigenetic regulation during FBAT-development in rabbits, miRNA-seq was carried out to determine the miRNA expression profiles and changes from G21 to G24. An average of 6.9 million clean reads were obtained from all miRNA-seq libraries, and the Q30 values of the clean reads were 100% in all samples (Appendix A). The length distribution of the clean reads showed that the lengths of most reads ranged from 18 to36 bp, with the majority being 22 bp (Figure 3A). A total of 1097 miRNAs were expressed in the samples, including 533 known and 564 novel ones. The G21 and G24 shared 876 expressed miRNAs (Figure 3B and Appendix A). Hierarchical clustering of samples using the CPM values of the miRNAs showed samples in G21 and G24 formed a major cluster, respectively (Figure 3C). Differential expression analysis identified 57 and 53 known miRNAs significantly up- and downregulated (|log2(fold-change)| > 1 and *p*-value < 0.05) from G21 to G24, respectively (Figure 3D and Appendix A). According to the *p*-values, the top 10 significantly upregulated known miRNAs were miR-378-3p, miR-211-5p, miR-378-5p, miR-29b-3p, miR-29c-3p, miR-365-3p, miR-144-5p, miR-144-3p, miR-195-5p, and miR-30e-5p. The top 10 significantly downregulated known miRNAs were miR-9-3p, miR-214-3p, miR-335-3p, miR-137-3p, miR-100-5p, miR-496-3p, miR-377-3p, miR-433-3p, miR-214-5p, and miR-133b-5p. On the other hand, 77 and 76 novel miRNAs were up- and downregulated from G21 to G24, respectively (Figure 3D, Appendix A). These data reveal the changes of miRNAs during the early development of rabbit FBATs.

### 3.3. Chromatin Accessibility Changes during the Early FBAT Development in Rabbits

To detect the chromatin-level epigenetic transcriptional regulation, we next carried out ATAC-seq to detect genome-wide chromatin accessibility. Three and four chromatin pools from G21 (*n* = three) and G24 (*n* = four) were subjected to ATAC-seq, respectively. We obtained an average of 120.83 million quantified reads and a 93.69% mapping ratio per sample (Appendix A). All the libraries showed expected fragment length, containing a higher proportion of nucleosome-free fragments and spanning fragments of mononucleosome (Appendix A). Distribution of the average ATAC-seq signals across all genes showed that there were strong signals present around the transcription start sites (TSSs, Figure 4A). In total, 131,739 and 130,561 chromatin-accessible regions (or peaks) were identified in G21 samples and G24 samples, respectively (Figure 4B), and 167,453 peaks were identified in all samples using DiffBind (Appendix A). Hierarchical clustering of the samples using normalized read counts (CPM values) of genome-wide peaks showed excellent reproducibility among replicates (Figure 4C). Genomic annotation of the peaks showed that 52.37% of peaks were located in the intergenic regions of genes, 31.23% in introns, 14.21% in promoters (±3 kb of TSSs), 1.74% in exons, and 0.44% in UTRs (Figure 4D).

To detect the changes of chromatin accessibility during the early development of FBATs, a differential analysis of the peaks was performed. A total of 7103 increased and 9843 decreased peaks were identified from G21 to G24 (Figure 4E). The highly accessible peaks (log(CPM) > 5) were broadly increased from G21 to G24 (202 increased vs. 40 decreased peaks). Functional annotation of the nearby genes of these peaks showed that these genes were significantly enriched GO-BP terms involved in enzyme activities, such as transferase activity, positive regulation of transferase activity, and regulation of protein kinase activity (Figure 4F). For the DNA sequences of increased peaks from G21 to G24, the top 10 significantly enriched TF binding motifs were known as binding sites of EBF2, CEBPB, NF1-halfsite, HLF, NFIL3, EBF1, RXR, PPARE, ERRA, and NR2F1 (Figure 4G). For the DNA sequences of decreased peaks from G21 to G24, the top 10 significantly enriched TF binding motifs were known as binding sites of CJUN, ATF2, CREB5, ATF7, JUND, TGA5, MOYG, ATF1, TGA4, and TCF21 (Appendix A).

### 3.4. Proximal Regulation of Chromatin Accessibility and the miRNA Regulation during the Early Development of FBAT

To investigate the chromatin accessibility-regulating gene transcription, we first analyzed the relationship between peak intensity and expression of corresponding nearby genes. Correlation analysis (Pearson’s correlation coefficient) showed that different types of peaks had distinct effects on regulating nearby genes. The peaks in the intron, exons, and intergenic regions showed the lowest effect in regulating gene expression (intron: R = 0.06; exon: R = 0.05; intergenic regions: R = 0.05). The peaks in the 5′UTRs and downstream regions showed a moderate effect in regulating gene expression (5′UTRs: R = 0.17; downstream: R = 0.26). Meanwhile, the peaks in the promoters showed the highest effect in regulating gene expression (R = 0.33, *p*-value < 0.01) (Figure 5A). Importantly, increased chromatin accessibility was found in the promoter regions of the upregulated adipogenesis genes, such as *LPL*, *FABP4*, *CEBPA*, and *ADIPOQ*, and the upregulated thermogenesis genes *UCP1* and *CIDEA* (Figure 5B). The correlation analyses between chromatin accessibility and gene expression suggested the importance of chromatin accessibility in the promoter regions in proximal regulation.

In this study, we further comprehensively explored the changes of open chromatin in the promoter regions. Our data shows that 9655 genes have no peak (*n* peak = 0), 10,461 genes contain a single peak (*n* peak = 1), and 5406 genes contain multiple peaks (*n* peak > 1) in their promoter regions (Figure 5C). Most genes containing 0 peaks showed unexpressed or low expression, while most genes containing peak(s) showed higher expression (Appendix A). For the genes containing a single peak in the promoter, 2100 upregulated genes had an increased promoter peak (Appendix A; for example, *FABP4* in Figure 5B) from G21 to G24, 1898 downregulated genes had a decreased promoter peak (for example, *HSP90B1* in Appendix A), 919 upregulated genes had a decreased promoter peak (for example, *MPZ* in Appendix A), 3757 downregulated genes had increased promoter peak (for example, *REXO2* in Appendix A), and 1787 unexpressed genes had a promoter peak (for example, *CSTL1* in Appendix A). Similarly, for the genes containing multiple promoter peaks, 917 upregulated genes had peaks that were all increased, 432 downregulated genes had peaks that were all decreased, 227 upregulated genes had peaks that were all decreased, 1195 downregulated genes had peaks that were all increased, 566 non-expressed genes had multiple peaks, and 2069 gene had both increased and decreased peaks (Appendix A). Analysis of the major promoter peak of genes found that 470 upregulated ones had a major increased peak, 590 downregulated ones had a major decreased peak, 213 upregulated ones had a major decreased peak, and 796 downregulated ones had a major increased peak (Appendix A).

To learn the functions of chromatin accessibility-regulating gene expression, we then performed GO-BP enrichment and KEGG pathway analyses of the upregulated DEGs that had increased promoter peaks. The GO enrichment showed that the carboxylic acid metabolic process, oxoacid metabolic process, and organic acid metabolic process were the top three significantly enriched GO-BP terms by these genes. The KEGG pathway analysis showed that carbon metabolism, citrate cycle, and PPAR signaling pathway were the top three significantly enriched pathways (Appendix A). On the other hand, the pathways involved in energy metabolism were also significantly enriched by these genes, such as thermogenesis, fatty acid metabolism, and fatty acid degradation (Appendix A, Appendix A).

Furthermore, we analyzed whether miRNA targeted protein-coding genes (PCGs) that contained peaks in promoters. Our results show that a total of 816 PCGs were predicted to be regulated by both the chromatin accessibility in promoter regions and the miRNAs (Figure 5D). There are fewer PCGs that contained a decreased peak (from G21 to G24) in promoters that were targeted by miRNAs. The upregulated PCGs that have an increased peak in promoters were more frequently targeted by miRNAs, with a higher proportion of upregulated miRNAs than downregulated miRNAs, which indicates that the effects of miRNA-mediated negative regulation were lower than those of chromatin accessibility-mediated positive regulation of these PCGs. Notably, the downregulated PCGs (*n* = 200) that have an increased peak were also frequently targeted by miRNAs, with a higher number of upregulated miRNAs than downregulated miRNAs, which indicates that the effects of miRNA-mediated negative regulation were higher than those of chromatin accessibility-mediated positive regulation for these PCGs. KEGG pathway analysis showed these downregulated genes were involved in the apelin, insulin, and endocytosis signaling pathways.

### 3.5. Genome-Wide Footprinting Analysis Revealed EBF1 Is Important for the Early Development of FBAT

DNA sequences directly occupied by DNA-binding proteins are protected from transposition during library construction in ATAC-seq, resulting in thd deletion of ATAC-seq signals (so-called sequence “footprint”) in peaks. To further identify key TFs involved in FBAT development, we performed a TF footprinting analysis for the obtained ATAC-seq reads, which enabled us to learn the potential genome-wide binding events of TFs. For the 765 individual vertebrate TFs that were subjected to the TF footprinting analysis, 41 and 36 TFs were found to significantly increase and decrease [all TFs with −log10 (*p*-value) above the 95% quantile or differential binding scores smaller/larger than the 5% and 95% quantiles (top 5% in each direction) were considered differential binding TFs] their footprints from G21 to G24 across genome-wide peaks, respectively (Figure 6A). The top five TFs with increased footprint changes (binding score changes) were EBF1, EBF3, EBF2, CTCF, and NFIB (Figure 6B). Their illustrated visible change of footprints across genome-wide peaks showed deeper footprints in G24 compared that in G21, indicating that the TFs with increased footprints were possible functional regulators of the early FBATs development of rabbits. The top five TFs with decreased footprint changes were ZNF211, CUX2, CUX1, ZNF8, and POU3F1. Their illustrated visible changes showed shallower footprints in G24 samples, compared to that of G21 (Appendix A, Appendix A).

The analysis of the RNA-seq data found 390 TF genes expressed in our samples, among which 27 and 32 TF genes were significantly up- and downregulated (|log2(fold-change)| > 1 and Padj < 0.01) from G21 to G24, respectively (Appendix A). The significantly upregulated TF genes contained many known adipogenic TFs, such as EBF1, CEBPB, and PPARG. As a master brown adipogenesis TF, EBF2 was moderately upregulated with log2(fold-change) = 0.83 and Padj = 0.077. Integrated analysis of gene expression changes and footprint changes of TFs found that 18 TFs had significantly changed gene expressions and significantly changed footprint changes across genome-wide open chromatin regions from G21 to G24. Among the 18 TFs, BHLHE22, PAX7, PAX3, ARX, CUX2, POU3F4, and POU3F3 were not only significantly downregulated (|log2(fold-change)| < −1 and Padj < 0.01) gene expression levels from G21 to G24, but also significantly decreased [−log10(*p*-value) above the 95% quantile or differential binding scores smaller than the 5% quantile] footprints from G21 to G24 (Figure 6C). For each of the seven TFs, the number of their downregulated [log2(fold-change) < 0] bound genes was more than that of the upregulated [log2(fold-change) > 0] bound genes from G21 to G24, indicating their potential TF regulation (Figure 6C). On the contrary, CEBPB, FOXN1, and EBF1 were not only significantly upregulated (|log2(fold-change)| > 1 and Padj < 0.01) gene expression levels from G21 to G24, but also significantly increased [-log10(*p*-value) above the 95% quantile or differential binding scores larger than the 95% quantile] footprints from G21 to G24, among which EBF1 had the highest gene expression level and the highest ratio of upregulated bound genes (Figure 6C). Notably, the binding motif of EBF1 was also significantly enriched by sequences of the increased peaks in this study (Figure 4G), showing its key role during early FBAT development in rabbits.

Although our data shows that EBF1 is an important TF involved in the early development of FBAT development, there is still a poor understanding of how EBF1 regulates its primary downstream targets, and consequently its secondary targets, during early FBAT development. We therefore performed the cascaded binding prediction of EBF1. For instance, EBF1 was predicted to bind to the promoter of its primary target-TF gene *THRA* to increase *THRA* expression [log2(fold-change) = 0.44 and Padj = 0.001], then the protein of upregulated *THRA* was predicted to bind to the promoter of EBF1′s secondary target-TF gene *NR4A1* to increase *NR4A1* expression ((log2(fold-change) = 1.92 and Padj = 0.0002, Figure 6D). Thus, we searched all upregulated [log2(fold-change) > 0] primary target-TFs and upregulated [log2(fold-change) > 0] secondary target-TFs of EBF1 and established an EBF1-regulated TF network (Figure 6E). The cascade network contained 11 primary target-TFs and 31 secondary target-TFs of EBF1. PPARG, EBF1, CEBPB, RXRG, and BCL6 were the top five upregulated DEG-coded TFs with the highest gene expression levels, their upregulated gene expression levels were validated by RT-qPCR (Appendix A). GO-BP enrichment of the TF network indicated that the predicted EBF1-regulated genes (including both TF genes and non-TF genes) were significantly enriched in lipid metabolisms, such as the top three enriched GO-BP terms of lipid storage, lipid localization, and carboxylic acid metabolic process (Figure 6F). KEGG pathway analysis showed that carbon metabolism, citrate cycle, and PPAR signaling pathway were the top three significantly enriched pathways, suggesting the important role of EBF1-regulated network in regulating FBAT development (Figure 6F).

In order to investigate whether miRNAs were involved in the EBF1-regulated gene network, we integrated the results of the miRNA target prediction (which were conducted using both miRanda [40] and RNAhybrid [41]) and the expression of EBF1-regulated genes. A total of 14 miRNAs were predicted to target 28 node genes of EBF1-regulated gene network (Appendix A), among which miR-329-5p, miR-125b-3p, miR-370-5p, miR-873-5p, and miR-214-3p were the top 5 miRNAs that were significantly downregulated from G21 to G24, potentially releasing the inhibition of node genes in EBF1-regulated gene networks. The results of RT-qPCR validated the downregulation of the top five miRNAs (Appendix A). GO-BP enrichment found that the genes targeted by the 14 miRNAs were significantly enriched in the GO-BP terms of fatty acid metabolic process, autophagy, and the process-utilizing autophagic mechanism. The top three significantly enriched KEGG pathways were mitophagy, pentose phosphate pathway, and fatty acid elongation. (Appendix A).

## 4. Discussion

In contrast to mice and humans, the molecular mechanisms responsible for BAT development, and especially for FBAT development, remain poorly understood in domestic animals. In the current study, changes in cell morphology were determined during the early development of FBAT. Our results show that both G21 and G24 samples contain cells that are composed of multilocular adipocytes, which are similar to the classical brown adipocytes found in humans [52]. The ascending number of multilocular adipocytes and the increased cell sizes and lipid accumulation of the adipocytes indicate that the period from G21 to G24 is important for FBAT development in rabbits. The dramatically elevated gene and protein expression levels of the thermogenic genes suggest the increased capacity of uncoupling respiration from G21 to G24. The changes of genome-wide gene expression show that the upregulated genes are significantly enriched in GO-BP terms and KEGG signal pathways involved in oxidative metabolisms and lipid metabolic process, reflecting the function formation of mature brown adipocytes of rabbit FBAT at G24. The PPAR signaling pathway plays a key role in adipocyte differentiation [53]. The upregulated genes were significantly enriched in the PPAR signaling pathway, indicating its key role in the early development of FBAT in rabbits. The downregulated genes were significantly enriched in the GO-BP terms and KEGG signal pathways involved in cell proliferation, which might suggest that the speed of proliferation of cells decreases during differentiation and lipid accumulation of fetal brown adipocytes.

Understanding the gene transcriptional networks is a long-standing challenge. MiRNAs are one of the epigenetics regulators that constitute a class of short ncRNAs, regulating gene expression by targeting 3′UTRs of messenger RNA [54]. MiRNA is highly conserved among species. Previous in vitro studies identified a group of miRNAs or miRNA families involved in regulating brown fat development in mice [55,56,57]. Our results show that the top 10 upregulated miRNAs overlap with the members of the miRNA families that promote brown fat development in mice, such as miR-378s, miR-365-3p, miR-195-5p, and miR-30e-5p, which demonstrates that these miRNAs might promote brown adipocyte differentiation in both intrauterine and extrauterine environments. On the other hand, we identified novel DEmiRNA from our samples, but their function and regulatory mechanism during FBAT development need future investigation.

Recent studies revealed that chromatin accessibility is a critical epigenetics factor regulating tissue development in pigs [58], chickens [59], and cattle [59]. The unfolding DNA sequences around nucleosomes in eukaryotes form active regions, such as promoters and enhancers, thus affecting downstream gene expression [60]. As revealed by ATAC-seq analysis, the ATAC-seq signals have been found to be enriched around the TSS regions of genes in many previous studies, as well as in this study [58,61,62]. The majority of peaks have been mapped to intergenic regions, introns, and promoters, while very few peaks were mapped to exon and UTR, of which peak distribution was similar to previous ATAC-seq studies [58,63]. In this study, the number of highly accessible peaks in G24 is more than that in G21, and nearby genes of these increased peaks are involved in enzyme activities, which suggests that highly accessible peaks might be important for maintaining basal functions and activities in mature brown adipocytes. Combined analysis of chromatin accessibility and transcriptome data shows that obviously increased peaks were found in the promoters of some key upregulated adipogenesis genes and thermogenic genes, indicating the importance of the chromatin accessibility in gene promoters for gaining BAT properties. Promoter-based combined analysis of chromatin accessibility and transcriptome revealed that some genes are susceptible to chromatin accessibility of gene promoter regions (e.g., upregulated gene had an increased promoter peak), while other genes are not susceptible to chromatin accessibility of gene promoter regions (e.g., upregulated gene had a decreased promoter peak). Here, we consider several possible reasons for this observation: (1) the chromatin accessibility is not sufficient for gene expression, as chromatin accessibility only provides the probability of TF binding, (2) the DNA sequence of a peak(s) of a promoter could bind to different TFs, while the transcriptional effect of TFs is affected by the combination of these TFs and the abundance of TFs, (3) some genes are simultaneously regulated by a proximal promoter and a distal enhancer, and (4) the transcriptional effect of chromatin accessibility is countered by other epigenetics factors, such as DNA methylation, histone modification, and non-coding RNAs, in this study. Interestingly, the GO-BP enrichment and KEGG analysis for the upregulated genes that had increased promoter peaks show that these genes are broadly involved in lipid-associated biological processes and pathways, such as lipid metabolic process, adaptive thermogenesis, citrate cycle, and the PPAR signaling pathway, which might suggest that proximal regulation of chromatin accessibility plays key roles during the early development of FBAT.

Footprinting analysis of ATAC-seq data enables the investigation of transcription factor (TF) binding, and has become a powerful tool for finding key TFs involved in the various biological process [48,50,62]. EBF1 was previously considered a TF involved in white adipocyte differentiation [64]. A recent study identified EBF1 as a candidate regulator of the cold response in BAT [65]. Deletion of EBF1 alone in adipocytes had minor effects on BAT, while concurrent deletion of EBF1 and EBF2 caused a complete loss of brown fat thermogenic fate and ablated BAT recruitment during chronic cold exposure (4 °C) [65]. Our results shows that both EBF1 and EBF2 significantly increase their TF footprints from G21 to G24. On the other hand, the upregulation extent of EBF1 is found more than EBF2 in gene expression levels, which might indicate that EBF1 plays crucial role in the early development of FBAT by increasing both gene expression levels and downstream gene binding at a constant temperature (approximately 37 °C) in utero. In this study, we combined the degrees of gene expression level, gene change degree, and footprint change of TFs identified EBF1 as a key TF involved in FBAT development and revealed an EBF1-regulated two-layer cascade-regulating network. As revealed by GO enrichment, the EBF1-regulated gene network is significantly enriched in lipid metabolism, which suggests that the EBF1 might promote FBAT development by directly binding primary target genes or regulating secondary target genes that are involved in adipogenesis. Furthermore, we reveal that several miRNAs regulate the early development of FBAT by targeting the genes involved in the fatty acid metabolic process, suggesting that the EBF1-regulated network might be regulated by post-transcriptional mechanisms. Thus, the specific interaction and functional impact of the identified EBF1-regulated TF networks and miRNAs warrant further in vitro and in vivo validation.

## 5. Conclusions

In conclusion, this study represents the first to describe the histological, transcriptome, miRNAs, and chromatin accessibility of the early development of FBAT in rabbits. Both miRNAs and chromatin accessibility are identified as potential epigenetic factors for FBAT development, and the genes that tend to be regulated by chromatin accessibility and miRNAs are identified, respectively. Moreover, combined analyses predict an EBF1-regulated cascade TF-network, which plays an important role in regulating FBAT development. Our work provides a framework for understanding epigenetics regulatory mechanisms during the early FBAT development and identifies potential TF involved in the early development of FBAT in rabbits. Therefore, these data can be used for future research on the BAT-based thermogenic capacity of domestic animals.

## Figures and Tables

**Figure 1 cells-11-02675-f001:**
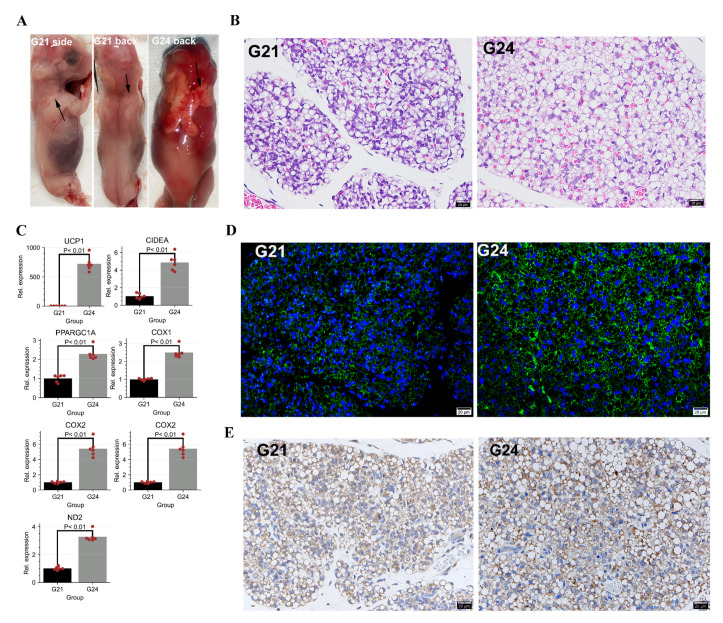
Histological analyses of the early development of fetal BATs (FBATs). (**A**) Rabbit FBATs in gestational day 21 (G21) and gestational day 24 (G24). The arrows show the FBAT depots. (**B**) H&E staining of rabbit FBATs. The scale bars: 20 μm. The blue signals show the cell nucleus. (**C**) RT-qPCR detects mRNA expression levels of UCP1, CIDEA, PPARGC1A, COX1, COX2, ND1, and ND2. The bar shows the means of 6 independent experiments. Two technical replicates were set for one individual experimental replicate. The “Rel.” represents “Relative”. The expression is normalized to the RN18S and G21. Each red dot shows the relative expression level of one experimental replicate. (**D**,**E**) Uncoupling protein 1 (UCP1) immunofluorescence (IF) and immunohistochemical (IHC) of rabbit FBATs. The scale bars: 20μm. The blue signals show the nucleus. The green signals and brown signals show the expressed UCP1 proteins in IF and IHC, respectively.

**Figure 2 cells-11-02675-f002:**
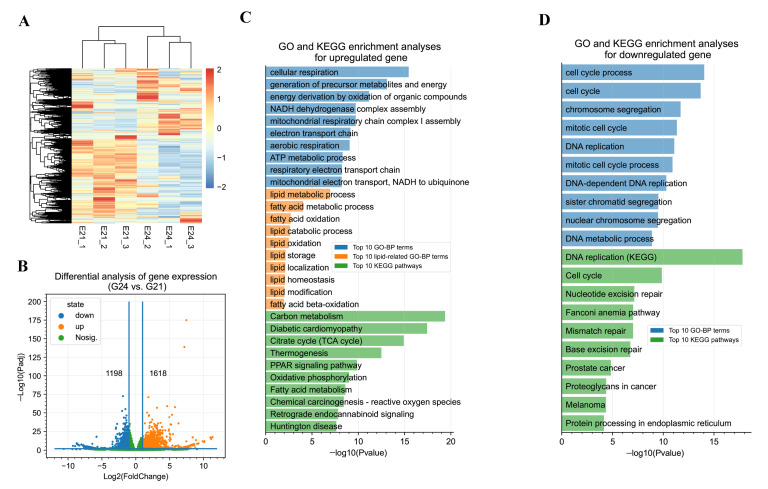
Deep RNA-seq reveals transcriptome changes during the early development of fetal BATs (FBATs). (**A**) Heatmap analysis and hierarchical clustering of samples using normalized read counts (TPM values) of the transcriptome. Each row shows one gene, and the TPM values were Z-scaled by row. The scale bar shows the Z-scaled TPM values. (**B**) Volcano plot showing significantly up- and downregulated genes from G21 to G24. (**C**) GO and KEGG analyses of upregulated genes. (**D**) GO and KEGG analyses of downregulated genes.

**Figure 3 cells-11-02675-f003:**
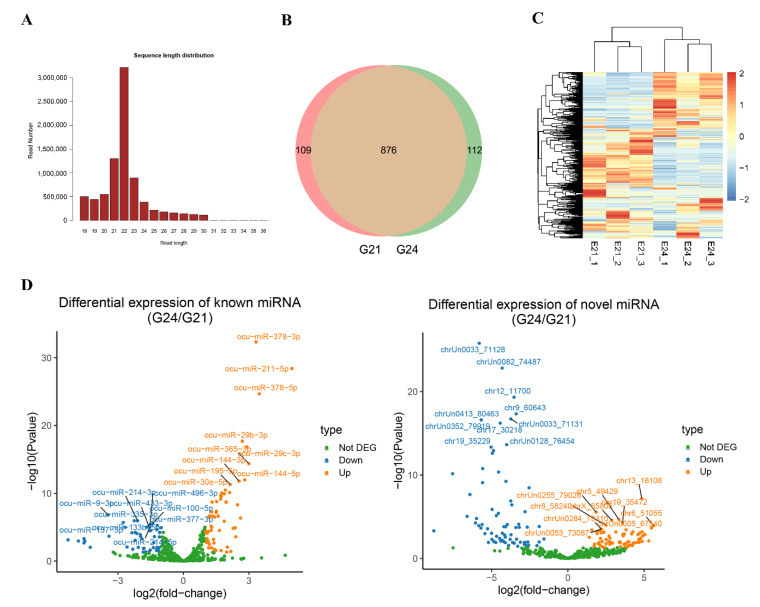
MiRNA-seq reveals the miRNA profiles and changes during early FBAT development. (**A**) Length distribution of reads in a representative miRNA-seq library. (**B**) Venn diagram showing the number of expressed miRNAs in G21 and G24. (**C**) Heatmap analysis and hierarchical clustering of samples using the CPM values of miRNAs. Each row shows one miRNA, and the CPM values were Z-scaled by row. The scale bar shows the Z-scaled CPM values. (**D**) Differential expression analysis of known miRNAs (left panel) and novel miRNAs (right panel). The top 10 up- and downregulated miRNAs are shown.

**Figure 4 cells-11-02675-f004:**
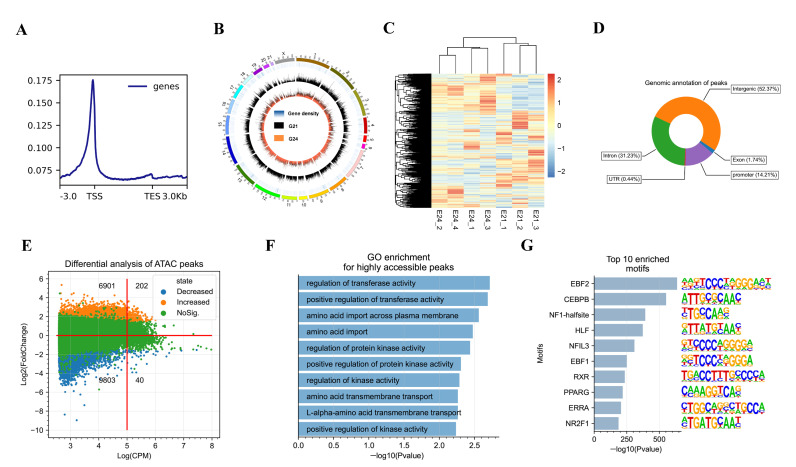
Landscape and changes of chromatin accessibility during the early FBAT development in rabbits. (**A**) The distribution plot of sequencing reads from a representative ATAC-seq library across all genes. (**B**) Genome-wide chromatin accessibility of rabbit FBATs. (**C**) Hierarchical clustering of samples using normalized read counts (CPM values) of genome-wide peaks. (**D**) Genomic annotation of identified ATAC-seq peaks. (**E**) MA plot of differential peaks. The number showed the differential peaks in the corresponding quadrant. (**F**) The top 10 enriched GO-BP terms by the nearby genes of increased highly accessible peaks. (**G**) The top 10 significantly enriched TF binding motifs by increased peaks from G21 to G24 according to the enrichment *p*-values. The corresponding binding motif of TFs are shown.

**Figure 5 cells-11-02675-f005:**
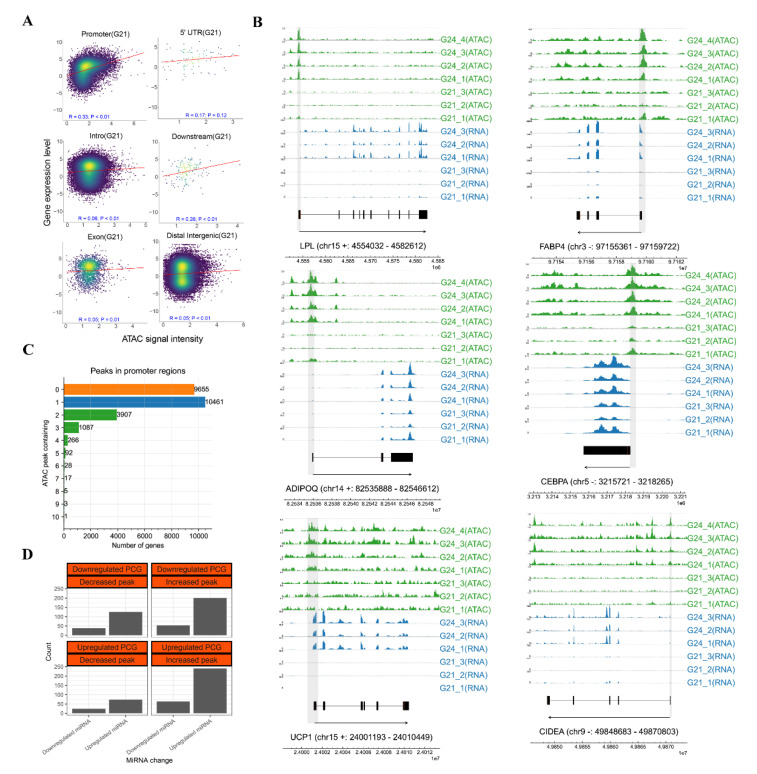
Integrated analysis of chromatin accessibility, miRNAs, and gene expression during FBAT development in rabbits. (**A**) Gene element-based correlation analyses between peak intensity and gene expression. The *X*-axis represents the log-scaled CPM values of the ATAC-seq peaks and the *Y*-axis represents the log-scaled TPM values of the corresponding nearby genes. R is Pearson’s correlation coefficient and P is the significance test for the correlation coefficient. (**B**) ATAC-seq (green) and RNA-seq (blue) tracks of key BAT genes. Gene exons are marked in black. Gene information is marked by “gene name (chromosome strand: start–end)”. The location of differential peaks is shadowed in grey. See also in the below tracks. (**C**) The number of genes containing different numbers of peaks in the promoter regions. The orange, blue, and green bars show the zero peak, single peaks, and double/multiple peaks for a gene, respectively. (**D**) Numbers of genes that were regulated by both chromatin accessibility and miRNAs.

**Figure 6 cells-11-02675-f006:**
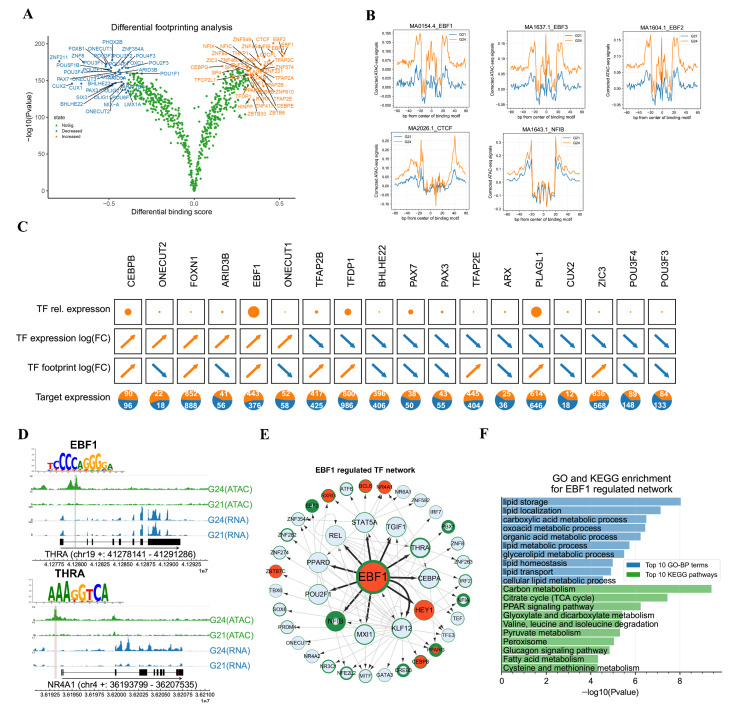
TF footprinting analysis identified the key TFs involved in the early development of FBAT. (**A**) Volcano plot showing the comparison of TF footprint between G21 and G24 across genome-wide ATAC-seq peaks. (**B**) Aggregated footprint plot of the top 5 TFs that increased their footprints. An aggregated footprint for all associated transcription factor binding sites is shown for each TF. Individual plots are centered around binding motifs. (**C**) Integrative analysis of relative expression, expression fold-change, footprint fold-change, and expression of the bound gene of key TFs during fetal BAT development. The area of the circle shows the relative expression of TF genes. An arrow pointing to the upper right corner shows an upregulated TF gene, or the increased footprint of one TF from G21 to G24. An arrow pointing to the lower right corner shows a downregulated TF gene, or the decreased footprint of one TF from G21 to G24. The orange and blue parts of a pie plot show the ratio of the upregulated bound genes and the downregulated bound genes of one TF, respectively. The numbers of the bound genes are marked in the corresponding part of the pie plots. (**D**) One path of the EBF1-regulated TF cascade network. The tracks show one path of the EBF1-regulated cascade network. The upper tracks show EBF1 binding to the promoter region of THRA and promoting THRA expression. The bottom tracks show the THRA binding to the promoter of NR4A1. The location of the peak is shadowed by grey color. The location of the binding site is shadowed by red color. The binding motifs of EBF1 and THRA are shown at corresponding bind sites. (**E**) EBF1-regulated two-layer TF cascade network. Sizes of nodes represent the level of the network, starting with EBF1 (Large: EBF1, Medium: 1st level, Small: 2nd level). Nodes are colored based on the corresponding fold-change of gene expression (G24 vs. G21). The thickness of the node border shows the relative expression of TF genes. Directed edges indicate binding sites in the respective gene promoter. (**F**) GO and KEGG analyses of genes in the EBF1-regulated cascade network.

## Data Availability

The datasets generated for this study can be found in the Sequence Read Archive (https://www.ncbi.nlm.nih.gov/sra, accessed on 8 February 2022) at NCBI, with the BioProject ID: PRJNA870319.

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
