# Peer review of "Integrated Analysis of Transcriptome, microRNAs, and Chromatin Accessibility Revealed Potential Early B-Cell Factor1-Regulated Transcriptional Networks during the Early Development of Fetal Brown Adipose Tissues in Rabbits"

_cells, 2022, doi:10.3390/cells11172675_

Round 1
Reviewer 1 Report
In the manuscript 'Integrated analysis of transcriptome, microRNAs, and chromatin accessibility revealed potential early B-cell factor1-regulated transcriptional networks during the early development of fetal brown adipose tissues in rabbits', Du and co-authors described the transcriptional/chromatin accessibility changes during a key developmental stage of brown adipose tissue maturation. It is an important resource for the brown fat field.
It would be more informative, if the cellular composition could be resolved by de-convolute the bulk RNAseq data with public available single cell RNAseq data of BAT.
Author Response
Reviewer #1
- It would be more informative, if the cellular composition could be resolved by de-convolute the bulk RNAseq data with public available single cell RNAseq data of BAT.
>>> We thank the reviewer for this important suggestion. Single-cell RNA-seq (scRNA-seq) is a robust tool to analyze for dissecting cellular heterogeneity, and our data showed that cellular composition changed during the early development of FBAT. However, currently, the opportunity to use public data to assist in the analysis of fetal brown adipose tissue (FBAT) development remains limited, mainly because (1) There was no report for brown adipose tissue (BAT) or FBAT in single-cell resolution in rabbits, (2) Cross-species analysis of scRNA-seq data of FBAT might be informative, but even for a universal model animal like the mouse, we lack scRNA-seq public data of FBAT. (3) Although there are several scRNA-seq reports for postnatal development of BAT in mice or other species, prenatal and postnatal development of BAT might not be comparable due to a great difference in developmental condition (e.g., constant temperature environment in the uterus vs. cold exposure during postnatal development).
Therefore, the research on FBAT in single-cell resolution was worth investigating. With the deepening of FBAT research and the decrease in the cost of scRNA-seq, deconstruction of FBAT development in single-cell resolution could be significant in further rabbit studies.
Reviewer 2 Report
In this manuscript, Du et al. provides a framework for understanding epigenetics regulatory mechanisms underlying the early development of fetal brown adipose tissue (FBAT) in rabbits, which are economically important domestic animals. FBAT is pivotal for newborns to maintain core temperature during the first several days of life.
Overall, this study is interesting because the authors combined several high-throughput approaches such as RNA-seq, miR-seq and ATAQ-seq to provide an integrative analysis enabling the identification of potential transcriptional factors involved in the early development of FBAT.
However, the manuscript suffers from major flaws in the present form. The authors must address some important points before the manuscript can be reconsidered for publication in Cells.
Major points:
1. The authors provided huge amount of data (RNA-seq, miR-seq and ATAQ-seq): the raw data must be submitted on GEO NCBI database.
3.
5. significantly down-regulated known miRNAs. Please precise if it is the adjusted p-value? And precise the fold-change for each miRNA and threshold for the adjusted p-value. Did they check whether these miRNAs belong to a particular cluster of miRNAs?
significantly down-regulated known miRNAs. Please precise if it was the adjusted p-value? And precise the fold-change for each miRNA and the threshold for the adjusted p-value. Did they check whether these miRNAs belong to a particular cluster of miRNAs?
7. please precise if it is the adjusted p-value? And precise the fold-change for each transcription factor and the threshold for the adjusted p-value.
the authors stated that the top 5 TFs with decrease footprint changers were ZNF211, CUX2, CUX1, ZNF8 and POU3F1. Please explain this statement because the reviewer was not convinced with this interpretation.
9. please precise the threshold for the adjusted p-value? And the fold-change for each gene and transcription factor.
1. PRDM16 also plays pivotal role in the control of BAT development. Did the authors check its expression in their RNA-seq analysis?
1. Angueira et al. already demonstrated that deletion of EBF1 alone in adipocytes had minor effects on BAT. However, concurrent deletion of EBF1 and EBF2 caused a complete loss of brown fat thermogenic fate and ablated BAT recruitment during chronic cold exposure (Cell Rep. 2020 March 03; 30(9): 2869–2878.e4). Did the authors check EBF2 expression in the RNA-seq analyses? This issue should be stated and discussed in the present manuscript.
Minor points:
The importance of BAT in humans has been published in 2009 in 3 manuscripts in the same NEJM issue. The authors have already cited Cypess et al. but
4. RT-qPCR should be used instead of qRT-PCR. Please check in the whole manuscript and in the supplemental data.
5. Table S1: please provide ID for CEBPB and RN18S
6. Figure 1B: please provide a higher magnification to better observe cell morphology and multilocular adipocytes. Same remark for Figures 1D and 1E.
7. Gene names should be written in lower case and italics. Please check.
8. Figures 1F, 2C, 3C: could the authors present heatmap illustrations instead of dendrograms of samples.
9. Figures 1F and 1G are blurred. Please improve the resolution of the illustration.
1. Line 466: miRNA target prediction software should be mentioned.
1. Line 431: Figure S4B appears to be unrelated to the statement.
1. Line 431-436: the sentence is not clear. Please clarify.
1. Line 438: CEPBB instead of CEPBA
1. Figure S4: please explain why some points have no SEM and precise the number of experimental determinations?
Author Response
Reviewer #2
Major points:
- The authors provided huge amount of data (RNA-seq, miR-seq and ATAQ-seq): the raw data must be submitted on GEO NCBI database
>>> We thank the reviewer for this important note. Based on the reviewer’s comment, we uploaded raw data of RNA-seq, miRNA-seq, and ATAC-seq to the NCBI SRA database (lines 663-664). To keep the data confidential before manuscript publication, we have set the temporary non-disclosure of data. Upon our manuscript publication, we will release these data in the SRA database. After publication, the reviewers and readers can access these data via accession number: PRJNA870319. We provide a screenshot of the SRA database to prove the uploading of datasets.
- Line 214: authors must clarify how they quantified miRNA expression? Did they use SYBR green or TaqMan probes for miRNA quantification? How was specificity verified for quantification of a specific miRNA? Please precise the complete procedure in the Materials and Methods section and eventually the ID of the Taqman probes used for each miRNA.
>>> We thank the reviewer for this important suggestion. We regret these missing statements in the Materials and Methods section. We used the SYBR method to quantify the miRNA expression using Mir-X miRNA First-Strand Synthesis Kit (Takara, Dalian, China) following the manufacturer’s instructions. Melting curve analysis was performed from 55 to 95 °C with increments of 0.5 °C, and the unique melting peak of the expected PCR product was used to validate the primer specificity. Please see our revised manuscript for details (lines 216-227). We used SYBR rather than the Taqman method for miRNA quantification, probes IDs were not available.
- Line 201 (RT-qPCR): SYBR green was used to quantify gene expression. Please precise how the specificity of primers was checked and precise the efficacy for each primer couple.
>>> We thank the reviewer for this comment. Melting curve analysis was performed from 55 to 95 °C with increments of 0.5 °C, and the unique melting peak of the expected PCR product was used to validate the primer specificity. Please see our revised manuscript for details (lines 211-214). For the efficacy of each primer couple, the Ct values were variable and mainly depended on the intrinsic gene expression. In general, one PCR reaction (including that in our study) with the Ct value < 35 was considered efficient amplification.
- Line 214 and 217: relative miRNA expression must be determined by the formula 2^-DDCt and not 2^DDCt. Please check and clarify since this mistake may change the interpretation of data.
>>> We thank the reviewer for this important note. We are sorry for the typing error. We change 2^DDCt to 2^-DDC in our revised manuscript. Please see our revised manuscript for details (lines 215 and 227).
- Line 289: the authors provided the top 10 significantly down-regulated known miRNAs. Please precise if it is the adjusted p-value? And precise the fold-change for each miRNA and threshold for the adjusted p-value. Did they check whether these miRNAs belong to a particular cluster of miRNAs?
>>> We thank the reviewer for this comment. The method of differential analysis of miRNA is often flexible in many studies. Maybe the number of genome-wide miRNAs was far less than that of mRNAs, the test of difference miRNA did not force to be adjusted when using multiple hypothesis tests, and many previous miRNA studies set the threshold using p-value rather than adjusted p-value (e.g., Zhou and colleagues in doi.org/10.1007/s10142-020-00763-8, Hailu and colleagues in doi.org/10.1038/s41390-021-01548-w, Wu and colleagues in doi.org/10.1186/s12870-021-03240-x, and Zhang and colleagues in doi.org/10.1038/srep22907). On the other hand, the number of known differentially expressed miRNAs was very small (57 upregulated and 53 downregulated). To increase the ratio of true positive differentially expressed miRNAs, we selected the p-value as the threshold for differential analysis of miRNAs, which was stated in the Materials and Methods section (line xx and line 154). In the revised manuscript, we also provided p-value and fold-change in the Result section. Please see our revised manuscript for details (line 302). Please understand us.
To learn whether miRNAs belong to a particular cluster of miRNAs, we retrieved miRNA family data (miFam.dat.gz) from miRbase. However, we found that there were no rabbit miRNAs annotated in any miRNA family. Therefore, further well-annotated miRNA family data could help us to know the more detailed miRNA clusters.
- Line 289: the authors provided the top 10 significantly down-regulated known miRNAs. Please precise if it was the adjusted p-value? And precise the fold-change for each miRNA and the threshold for the adjusted p-value. Did they check whether these miRNAs belong to a particular cluster of miRNAs?
>>> We thank the reviewer for this comment. This comment seems to be a repeat of comment 5.
- Line 418 (Figure 5A): please precise if it is the adjusted p-value? And precise the fold-change for each transcription factor and the threshold for the adjusted p-value.
>>> We thank the reviewer for this comment. As described in TOBIAS (Bentsen and colleagues in doi: 10.1038/s41467-020-18035-1), all TFs with -log10(p-value) above the 95% quantile or differential binding scores smaller/larger than the 5% and 95% quantiles (top 5% in each direction) were considered differential binding TFs. Our differential binding analysis of TF was performed completely according to the software instructions. The p-value need not be adjusted. In the revised manuscript, we also added the statement in the Result section. Please see our revised manuscript for details (lines 439 - 441).
- Figure S3A: the authors stated that the top 5 TFs with decrease footprint changers were ZNF211, CUX2, CUX1, ZNF8 and POU3F1. Please explain this statement because the reviewer was not convinced with this interpretation.
>>> We thank the reviewer for this note. The illustrated visible changes of the five TFs showed shallower footprints in G24 samples than in G21 samples, indicating the decreased footprints of the TFs. However, the order of footprint changes of the five TFs might be difficult to directly observe in these figures. Therefore, we provide the differential binding results of TFs in our revised manuscript in a Supplementary Table. The reviewers or readers could check the quantified footprint changes for each TFs. Please see our revised manuscript for details (Table S8).
- Line 433 (Figure 5C, D, E): please precise the threshold for the adjusted p-value? And the fold-change for each gene and transcription factor.
>>> We thank the reviewer for this note. We added the threshold for the statements in Figures 5C, D, and E. Please see our revised manuscript for details (lines 460, 466-467, 477, 479, 480, and 481).
- PRDM16 also plays pivotal role in the control of BAT development. Did the authors check its expression in their RNA-seq analysis?
>>> We thank the reviewer for this professional comment. PRDM16 is indeed an important gene in BAT development and were reported in many previous studies, especially for human and mice. However, the rabbit PRDM16 was not annotated in Ensembl. Because we used rabbit genome and gene annotation in Ensembl, we did not obtain any information on PRDM16 during data processing. Notably, the binding motif of PRDM16 currently is not in any database, which implied that it is difficult to conduct depth analysis of PRDM16 using our ATAC-seq data. Therefore, the resolution of PRDM16 is temporarily abandoned, and future PRDM16-related analysis might be conducted when rabbit PRDM16 is well-annotated.
- Angueira et al. already demonstrated that deletion of EBF1 alone in adipocytes had minor effects on BAT. However, concurrent deletion of EBF1 and EBF2 caused a complete loss of brown fat thermogenic fate and ablated BAT recruitment during chronic cold exposure (Cell Rep. 2020 March 03; 30(9): 2869–2878.e4). Did the authors check EBF2 expression in the RNA-seq analyses? This issue should be stated and discussed in the present manuscript.
>>> We thank the reviewer for this professional comment. The investigation of EBF1 conducted by Angueira et al. indeed showed a minor effect after deletion of EBF1 alone in adipocytes during brown adipocyte development, while concurrent deletion of EBF1 and EBF2 showed a great effect on the brown fat thermogenic fate and BAT recruitment during chronic cold exposure. In our study, the EBF2 was moderately upregulated with log2(fold-change) = 0.83 and Padj = 0.077 from G21 to G24. We presented the result of EBF2 and discussed it in our revised manuscript. Please see our revised manuscript for details (line 454 for the result and lines 596 - 603 for discussion).
Minor points:
- The importance of BAT in humans has been published in 2009 in 3 manuscripts in the same NEJM issue. The authors have already cited Cypess et al. but van Marken Lichtenbelt et al., and Schrauwen P et al. should be also cited.
>>> We thank the reviewer for this note. We cite van Marken Lichtenbelt et al., and Schrauwen P et al. in our revised manuscript. Please see our revised manuscript for details (line 52).
- Line 132: please precise if paired-end reads were generated for the RNA-seq analysis.
>>> We thank the reviewer for this note. The paired-end 150 bp reads were generated in RNA-seq in this study. We add this statement to our revised manuscript. Please see our revised manuscript for details (line 133).
- Line 155: p-values should be adjusted. Please clarify.
>>> We thank the reviewer for this suggestion. Please see the response in Major point 5, we had not adjusted p-values in miRNA-seq analysis.
- RT-qPCR should be used instead of qRT-PCR. Please check in the whole manuscript and in the supplemental data.
>>> We thank the reviewer for this note. We change the “qRT-PCR” to “RT-qPCR” in our revised manuscript and supplemental data. Please see our revised manuscript for details.
- Table S1: please provide ID for CEBPB and RN18S
>>> We thank the reviewer for this note and sorry for the negligence. We added ID for CEBPB and RN18S in our revised manuscript. Please see our revised manuscript for details (Table S1).
- Figure 1B: please provide a higher magnification to better observe cell morphology and multilocular adipocytes. Same remark for Figures 1D and 1E.
>>> We thank the reviewer for this note. We provided Figure 1, 1D, and 1E with higher magnification in our revised manuscript. Please see our revised manuscript for details (Figure 1).
- Gene names should be written in lower case and italics. Please check.
>>> We thank the reviewer for this comment. In general, the gene names in the lower case were applicable for mouse genes, while gene names in the upper case were applicable for other mammals (at least for humans and rabbits). Therefore, we kept gene names in upper case in our manuscript. Please understand us. On the other hand, as per the reviewer’s suggestion, gene names were written in italics. Notably, when gene names were expressed as proteins (e.g., TFs in this study), we used regular font. Please see our revised manuscript for details.
- Figures 1F, 2C, 3C: could the authors present heatmap illustrations instead of dendrograms of samples.
>>> We thank the reviewer for this note. The corresponding heatmap was presented in the revised manuscript. Please see our revised manuscript for details (Figure 2A, Figure 3C, and Figure 4C).
- Figures 1F and 1G are blurred. Please improve the resolution of the illustration.
>>> We thank the reviewer for this note. Zoomed Figures 1F and 1G were presented in the revised manuscript. Since we zoomed Figure 1B, 1D, 1E, 1F, and 1G, the size of the original Figure 1 was split into two major Figures to fit the publication requirements. Please see our revised manuscript for details (Figure 2A and 2B).
- Line 466: miRNA target prediction software should be mentioned.
>>> We thank the reviewer for this note. We added a software statement in the target prediction. Please see our revised manuscript (line 495).
- Line 431: Figure S4B appears to be unrelated to the statement.
>>> We thank the reviewer for this note and are sorry for this typing error. We revised Figure S4B to Figure S3B. Please see our revised manuscript (line 452).
- Line 431-436: the sentence is not clear. Please clarify.
>>> We thank the reviewer for this note. Because this sentence contained both TFs (protein) and TF corresponding genes statement, it seems not clear. We carefully rearranged the sentence. Please see our revised manuscript (lines 455 - 461).
- Line 438: CEPBB instead of CEPBA.
>>> We thank the reviewer for this note and are sorry for this typing error. We changed CEPBA to CEBPB in the revised manuscript. Please see our revised manuscript (line 465).
- Figure S4: please explain why some points have no SEM and precise the number of experimental determinations?
>>> We thank the reviewer for this comment. In this figure, the qRT-PCR results (solid lines) are expressed as the mean ±SEM of 3 independent experiments, and two technical replicates were set for one individual experimental replicate. However, the RNA-seq data (dashed lines) are expressed in RNA-seq fold-changes. Therefore, RNA-seq data in this figure have no SEM. The reviewers and readers can obtain a more detailed statement in figure legends of Figure S4 (lines 643-648).